# Enriched Motor Program [EMP]: Adaptation of a Physical Activity Intervention for Enhancing Executive Functions in Children with ASD

**DOI:** 10.3390/ijerph22060902

**Published:** 2025-06-05

**Authors:** Gabriele Gullo, Ambra Gentile, Marianna Alesi

**Affiliations:** Department of Psychology, Educational Science and Human Movement, University of Palermo, 90128 Palermo, Italy; gabriele.gullo01@unipa.it (G.G.); marianna.alesi@unipa.it (M.A.)

**Keywords:** physical activity, executive function, autism spectrum disorder, motor intervention program

## Abstract

Background: Recent studies indicate that physical activity (PA) may improve executive functions (EFs) in children with Autism Spectrum Disorder (ASD). The Enriched Motor Program (EMP), which combines aerobic and cognitive exercises, shows potential for enhancing EFs in these children. The EMP was originally created for typically developing preschoolers and includes locomotor and fine motor activities enriched by cognitive stimuli to help the development of EFs in children with ASD. The current study aims to adapt a shorter version of EMP for these children’s needs. Methods: The research will use a cross-sectional, quasi-experimental design with a forecasted sample of 40 children, with the age ranging from six to eight, with a diagnosis of ASD. The children’s working memory and inhibitory control will be measured before and after the intervention. Results: According to the literature, the experimental group should obtain higher scores, especially in working memory tasks. Discussion: This is the first implementation of EMP, which merges physical activities with cognitive stimuli to enhance EFs in children with ASD. It could be used by specialized centers and clinicians to support EFs through engaging activities, and it could be potentially recommended as a best practice for EF treatments in children with ASD.

## 1. Introduction

Physical activity (PA) is known to positively influence mental health and cognitive development in children and adolescents with neurotypical development [1,2] and those with neurodevelopmental disorders [3,4]. For example, benefits range from cognitive enhancement to independence and social inclusion and psychological well-being, including the improvement in self-esteem, self-competence, and self-efficacy [5]. When considering mental health and cognitive development in young people, it is fundamental to evaluate executive functions (EFs); this is a set of top-down cognitive processes that play a crucial role in cognitive and psychosocial development and support goal-directed behaviors by managing thoughts and actions to effectively and deliberately achieve objectives [6,7,8,9,10,11]. Researchers agree on considering three main functions [6]: inhibitory control, which is the ability to control impulses; working memory, which is the ability to mentally retain and elaborate information; and flexibility, which is the ability to adapt to the environment’s demands. Significant impairment in EF was observed both in children and youth with autism spectrum disorder (ASD), especially in hot EFs (e.g., social cognition, emotion regulation, and decision making) [12], flexibility, and working memory [13]. Therefore, it is crucial to improve EFs in children with ASD to reduce social and scholastic disparities [14] and enhance their adaptability to changing circumstances [6].

### 1.1. Summary of the Effectiveness of Physical Activity on Children with ASD

Many recent reviews and meta-analyses have highlighted the positive effect of physical activity (PA) in improving executive functions (EFs) in children with ASD [15,16,17,18]. However, the contrasting results of PA interventions appear to be influenced by several factors, such as the type, frequency, and duration of exercise [15,18]. Regarding the exercise type, a meta-analysis by Wu et al. [18] indicated that martial arts and aerobic exercises are notably effective in enhancing planning skills. Also, aerobic exercises significantly improved inhibitory control more than martial arts, motor skills programs, and no intervention. Additionally, a review by Grospetre et al. [15] showed that although sports and PA, in general, might be beneficial for EFs of children with ASD, some activities were revealed to be more effective than others for enhancing a specific cognitive function. Collective sports (e.g., football, basketball) and martial arts, which involve rapid decision making and high cognitive processing, seem to enhance cognitive functions more effectively, particularly concerning working memory [19] and cognitive flexibility [20]. Moreover, exergames, which combine PA and active video games, appear to be particularly effective for improving EFs in children with ASD [21] while improving motor skills [22]. Additionally, PAs with low motor skill involvement (e.g., stepping fitness and circuit exercises) were proven to still enhance executive functions [23,24]. As for the frequency of interventions, PA interventions with a frequency of three to seven times per week were shown to significantly enhance EFs, especially planning and inhibitory control [18]. In fact, it has been proven that increasing the frequency of interventions from one to a minimum of three times per week significantly amplifies the benefits [25]. Concerning the duration of interventions, recent reviews [15,18] identified a broad range, with an average protocol duration of twelve weeks. Consequently, Wu et al. [18] recommend implementing protocols lasting more than four weeks. The duration of each session should be considered as well, as the World Health Organization (WHO) recommends 150 min of moderate PA or 75 min of vigorous PA per week. However, it is difficult to determine the ideal duration of each session for children with ASD because of individual factors such as age, ability, and sensory needs [15].

Despite the absence of specific guidelines for PA intervention in ASD, recent meta-analyses [15,18] have established a foundation for designing motor programs tailored to the needs of children with ASD. Standardized clinical PA interventions should be conducted for a minimum of four weeks, at least twice a week. They should include aerobic exercises, which are particularly effective in enhancing executive functions, and collective activities, to reduce social dysfunctions.

### 1.2. Enriched Motor Program [EMP]: A Physical Activity Intervention for Enhancing Executive Functions in Children

EMP (original title: *PMA. Programma Motorio Arricchito* [26]) is a teacher-led program created to improve EFs (i.e., inhibitory control, working memory, cognitive flexibility, and planning) in typically developing preschoolers based on Diamond’s [27] and Diamond and Ling’s [28] recommendations, suggesting that complex and cognitively challenging movement tasks can result in EFs benefits. Particularly, EMP consisted of 30 units of locomotor and fine motor physical activities (e.g., throwing, catching, using tools like scissors) enriched by cognitive stimuli (e.g., Stroop with animals and fruit, word span, body parts span, a maze). Each program unit lasted about 60 min and included four phases: 1. warm-up (15 min), which included aerobic PAs like dribbling while walking and circular passing while jogging; 2. primary phase (25 min), focused on training motor skills and EFs; 3. cool down (10 min) for relaxing through a fun game; 4. a self-evaluation in which participants rated their performance using emoticons for happiness, indifference, or sadness.

EMP was found to effectively enhance children’s motor skills, behavior, and cognition, particularly in linguistic comprehension and expression, metacognition, and memory domains [26]. During the activity, children are stimulated to process cues and verbally explain their cognitive processes and personal experiences, leading to an improvement in linguistic abilities. Moreover, additional cognitive stimuli introduced during the activities have been proven to effectively increase EFs [29,30,31]. Exercises stimulating the association of complex cognitive skills and movements increased the speed and accuracy of performance on tasks requiring verbal and visuospatial working memory, flexibility, and inhibitory control abilities.

Based on previous studies demonstrating how conjugating aerobic exercises and cognitively engaging exercises are beneficial physiologically and cognitively to improve health in children with ASD, EMP could be a suitable program for enhancing motor skills and EFs in children with ASD, being beneficial physiologically and cognitively. The present work aims to adapt the EMP protocol to suit the cognitive and physical needs of children with ASD.

### 1.3. Objective and Hypotheses

The present work aims to present an adaptation of the EMP protocol to suit the cognitive and physical needs of children with ASD.

Specifically, we hypothesize that (H1) children in the experimental group would have higher EF scores in the post-test assessment than in the pre-test and (H2) children in the post-test group would have higher EF scores than the children in the control group.

## 2. Methods

### 2.1. Study Design

The present research protocol proposes a cross-sectional, quasi-experimental study design. Sample size is calculated with G*Power software (version 3.1.9.7) [32] and is determined considering a power of 0.80 and an effect size of d = 0.41, as reported by Wang, Cheng, and Li [19], which results in 36 children. Therefore, we will involve 40 children with ASD in case of participant loss. The research presents three phases:Pre-test evaluation (T0). This comprises four standardized tests to assess children’s working memory (two measures) and inhibitory control (two measures).Children will be randomly assigned to two groups through simple randomization. The experimental group will participate in the EMP activities twice a week for six weeks, while the other group will not engage in any activities or interventions.Post-test evaluation (T1). This comprises the same four standardized tests used in the pre-test evaluation phase and allows the assessment of children’s EFs after six weeks.

This research protocol has been approved by the Bioethical Committee of the University of Palermo with the protocol n° 173/2023 and will be conducted respecting the Declaration of Helsinki principles.

### 2.2. Sample

Children will be recruited through local centers and private specialists who work with children with ASD. The directors and specialists will manage the recruitment process. All parents will receive letters outlining the informed consent and will be asked to sign them if they agree to their children’s participation in the study. The signed consent letters will then be returned to the directors or specialists.

#### Inclusion Criteria

To be eligible for the study, children must meet following inclusion criteria: (1) to have received an ASD diagnosis; (2) to be between six and eight years old or have an equivalent mental age, calculated by the standardized test, The Correspondences and Functions Evaluation (which corresponds to Italian C.F.V.—Corrispondenze e Funzioni Valutazione) test [33]. This test has 42 items, subdivided into five areas of logical operations: qualitative correspondences, direct quantitative correspondences, indirect quantitative correspondences, direct functions, and indirect functions. Each item has a binary evaluation, with a mark of 1 for a correct answer and 0 for an incorrect answer. The raw data are transformed to a mental age (3–14 yrs) based on the appropriate conversion tables [33]; (3) to master language skills as early forms of understanding, listening, and expressing; pointing to body parts; following instructions, listening and paying attention; the use of names; sentence language; asking questions; and (4) to master motor skills including fundamental motor skills (FMS) of standing alone and walking, crawling, running, walking up/down stairs, and jumping, throwing, and catching. Language and motor skills will be measured through the VABS 2 -Vineland Adaptive Behavior Scales [34,35]. VABS 2 is the “gold standard” test to measure adaptive behaviors for everyday independent life from birth to 90 years old in intellectual or developmental disabilities, including autism spectrum disorders (ASD). It is a semi structured interview with the caregiver to measure skills in four areas: Communication (receptive, expressive, and written language), Daily Living Skills (skills to take care of oneself, and household and community skills), Socialization (social relationship, emotional, and behavioral regulation, leisure activities) and Motor Skills (fine and gross motor skills).

### 2.3. Description of Intervention

#### 2.3.1. Theoretical Framework of EMP

EMP was created based on four principles: simultaneous empowerment of EFs and motor skills, implementation of leisure activities, promotion of social and motivational competencies, and direct involvement of educators and teachers in delivering the program [26].

Physical activities can promote the development of EFs physiologically and psychologically. They enhance angiogenesis, as physical activities increase oxygenation of the prefrontal areas responsible for executive functioning and improve the metabolism and action of neurotransmitters in these areas [36]. Additionally, they promote mental health (e.g., regulation of the tone of humor, reduction in anxiety, etc.) and social inclusion [27,36,37].

Physical and leisure activities stimulate positive emotions, especially enjoyment [37]. Positive emotions tend to increase dopamine release in the frontal cortical areas, which, in turn, fosters executive functioning [38]. Physical activities also increase motivation to participate in the proposed activities, lowering the drop-out rates [26].

Finally, EMP can be implemented as part of educational programming over the long term by directly involving educators.

#### 2.3.2. EMP Activities Description

This research protocol proposes a short form of EMP, which comprises twelve units of the original protocol. The reduction from 30 to 12 activities considered the representativeness of the EFs and the nature of the motor exercises, i.e., gross coordination exercises. Therefore, according to Diamond’s model, equal numbers were selected for the three basic EFs: inhibitory control, working memory, and attentional shifting.

Each unit includes fifteen minutes of initial warm-up and thirty minutes of cognitively enriched activity. The initial warm-up session comprises dynamic stretch exercises (e.g., shoulder rolls, head circles, lungs) and aerobic exercises (e.g., running, jumping jacks). The following activity presents a physical activity exercise with a cognitive task derived from the neuropsychological literature and aimed to enhance one EF between working memory (five activities), inhibition (five activities), shifting (one activity), or planning (one activity). The activity will be repeated with variations to support cognitive function development. The program will be delivered in the form of group activities. The cognitively enriched activities and their variations are reported in Appendix A.

### 2.4. Measures

Each child will be tested individually in a session lasting approximately 10–15 min. The test will take place in a quiet room and will involve two tests assessing working memory capacity and two tests assessing inhibitory control, taken from the *FE-PS 2-6* battery for assessing EFs in children [39].

#### 2.4.1. Mr. Cucumber Test [40]

This test measures visual–spatial working memory and features an outline of a funny alien figure, to which stickers of various colors are attached. After a practice session, participants encounter three items at each level from 1 to 8, corresponding to the number of stickers in the 1 to 8 positions. Levels 1–5 are shown for 5 s each. From level 6, exposure time increases by one second per level. Participants then recall sticker positions on a colorless outline, pointing to the positions of the stickers, with the task focusing solely on recalling the positions. The test ends when a participant fails all three items at a level. They earn one point for each level where they correctly identify at least two items and one-third of a point for each correct item beyond that level. Participants are given an additional one-third of a point for each correct item beyond that level.

#### 2.4.2. Backward Word Span [41]

This test measures verbal working memory. Children are asked to repeat lists of semantically unrelated two- or three-syllable words in reverse order. The task begins with a practice trial (corrected if necessary) to demonstrate to the children how to repeat the word lists backward, followed by the test trials. Each list commences with three items, starting with two words and increasing until the participant fails to recall all three items at the same length, or until the maximum length of seven words is reached. The score is the number of words at the highest consecutive level of 23, which the participant repeated correctly in reverse order in at least two lists, plus one-third of a point for each correct list beyond that length. If a child fails to recall all three initial two-word lists backward but performed correctly in forward recall, a score of 1 is awarded. For a correct response on only one two-word list, the score is 1.33.

#### 2.4.3. Circle Drawing Task [39]

This test evaluates the ability to inhibit a continuous motor response. The task requires tracing a circle drawn on a white sheet of paper with a finger, adapting the execution speed to the examiner’s requests. Specifically, following a first execution in which the child is not given any indications regarding the speed of execution of the task, a second execution is planned in which they are explicitly asked to modulate their motor response, following the path as slowly as possible. The score corresponds to the difference between the second and the first execution’s time, converted into seconds.

#### 2.4.4. Day and Night Stroop Task [39]

This test measures inhibitory control. In the day and night Stroop task, the child must suppress the tendency to give a dominant response to a target. This is a monovalent task where only one feature (the sun or the moon) is presented on a card. The child must resist the inclination to provide a dominant response (e.g., saying “day” when presented with a card featuring the sun) in favor of a non-dominant response (saying “night” when a card with the sun is shown). This task requires both inhibitory control and working memory to recall the rule. However, the demands on working memory are modest and do not significantly affect performance [42]. The score is determined by the number of correct responses provided for the 16 stimuli.

### 2.5. Statistical Analysis

A repeated measures ANOVA will be run with the two groups (EMP vs. CG) × two-times points (Pre and Post) to assess both main and interaction effects. Specific effects will be evaluated performing post hoc tests, with the Bonferroni correction. Partial eta squared will be used to estimate the effect size, considering the following values: small (0.01 < η^2^_p_ ≤ 0.06), medium (0.06 < η²_p_ ≤ 0.14), and large (η^2^_p_ > 0.14).

## 3. Expected Results

We expect all scores to improve from the pre-test to the post-test (*p* < 0.05) only in the experimental group. Working memory is expected to show more consistent improvement than inhibitory control due to the cognitively engaging nature of EMP activities, which are particularly beneficial for enhancing working memory in children with ASD [15]. This activity might train specific aspects of executive functioning, particularly working memory, to store and update information [19]. This is particularly true for EMP activities, which conjugate collective physical activities with cognitive stimuli created ad hoc [26].

## 4. Discussions

Benefits from psychomotor interventions have been widely corroborated in children with autism spectrum disorders (ASD) because of their poor psychomotor skills and behavioral diseases. However, only a few studies have investigated the physical and cognitive outcomes of exercise programs in people with ASD.

Also in neurodevelopmental disorders, as in typical development, attempts have been made to plan and implement cognitively challenging motor programs to increase cognitive functions through motor exercises. These programs are expected to be efficacious in improving EFs because they challenge the abilities of suppressing incongruent stimuli (inhibitory control), manipulating complex sequences, and updating data (working memory), and adapting in real-time to contextual changes (cognitive flexibility).

To our knowledge, this is the first endeavor to implement a program that integrates physical activities with specially designed cognitive stimuli to improve executive functions in children with ASD. Furthermore, no previous attempts have been made to adapt the EMP, which has been shown to effectively enhance executive functions in children with typical development, to cater to the specific requirements of children with ASD. As a result, the research findings will support the program’s validation for this population.

## 5. Limitations

Despite its potential, this study has limitations. Recruiting a clinical sample is challenging, as it requires local clinical institutions and private specialists working with children with ASD. Another limitation is that the frequency of our intervention falls below the recommended three times per week, as suggested by the literature on EFs interventions. The reason is that most group interventions complementary to classic therapies, such as psychomotor skills and speech therapy, occur twice a week, as children with ASD typically participate in numerous other activities throughout the week (e.g., one-to-one cognitive-behavioral therapy). Finally, another possible limitation could be the compliance with this program if we consider the significant barrier characterizing the more general participation in Sport Intervention Programs, which is a sedentary lifestyle. Although the World Health Organization recommendations are at least 60 min of moderate to vigorous-intensity PA daily for ages ranging from 5 to 17 years, a large number of children with neurodevelopmental disorders do not meet these recommendations and show low levels of motivation to participate in regular exercise both at home and in educational settings.

## 6. Conclusions

If this research demonstrates the effectiveness of the EMP, it could be utilized by specialized centers and clinicians to support and enhance executive functions through enjoyable and engaging activities that can be easily administered.

Finally, it could be advocated as a “good practice” in clinical recommendations for EF treatments in children with ASD.

## Data Availability

Data will be available under formal request addressed to: comitato.bioetica@unipa.it.

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
