# Peer review of "Enriched Motor Program [EMP]: Adaptation of a Physical Activity Intervention for Enhancing Executive Functions in Children with ASD"

_ijerph, 2025, doi:10.3390/ijerph22060902_

Round 1
Reviewer 1 Report
Comments and Suggestions for Authors
Dear authors,
I recognize the relevance of the protocol and the adaptation of a Physical Activity Intervention for Children with Autism Spectrum Disorder. In my understanding the research protocol proposes an intervention based on a short form of EMP, which comprises twelve units of the original protocol and the measures to ascertain the impact of the intervention involve two tests assessing working memory capacity and two tests assessing inhibitory control. Despite the commendable effort of creating a PA protocol for children with ASD, overall, in my opinion the manuscript does not meet the scientific and technical standards required for publication in IJERPH.
My major concerns are:
- Reading throughout the paper it does not seems as a study protocol since, for instance, it is stated that: “Methods. The research uses a cross-sectional, quasi-experimental design with 40 children aged six to eight with ASD.”; … “It is expected that all scores will improve from pre-test to post-test in the experimental group”
- “2.2.1. Inclusion Criteria: To be eligible for the study, children must (1) have an ASD diagnosis; (2) be between six and eight years old or have an equivalent mental age; (3) be capable of communicating orally; and (4) have sufficient motor skills.” I have some concerns regarding those criteria, particularly how to assess “equivalent mental age” and the “sufficient motor skills”?
- The research protocol proposes an intervention based on a short form of EMP, which comprises twelve units of the original protocol. Are there adaptations? Did the authors test them with children with ASD? What was the rationale to choose those 12 units from the 30? Which of them?
- The measures to ascertain the impact of the intervention involve two tests assessing working memory capacity and two tests assessing inhibitory control. Is the FE-PS 2-6 battery for assessing EFs in children is validated for children with ASD?
- 167 – “The activity will be repeated with variations to support cognitive function development”. What type/kind of variations? If this is a paper to explain a protocol, every step and situation must be described.
- Limitations are not properly acknowledged in the paper.
Some reading recommendations:
Ataíde, S.S., Ferreira, J.P. & Campos, M.J. Prescription of Exercise Programs for Individuals with Autism Spectrum Disorder: Systematic Review. J Autism Dev Disord (2024). https://doi.org/10.1007/s10803-024-06566-1
Jia M, Zhang J, Pan J, Hu F and Zhu Z (2024) Benefits of exercise for children and adolescents with autism spectrum disorder: a systematic review and meta-analysis. Front. Psychiatry 15:1462601. doi: 10.3389/fpsyt.2024.1462601
Toscano CVA, Ferreira JP, Quinaud RT, Silva KMN, Carvalho HM, Gaspar JM. Exercise improves the social and behavioral skills of children and adolescent with autism spectrum disorders. Front Psychiatry. 2022 Dec 22;13:1027799. doi: 10.3389/fpsyt.2022.1027799. PMID: 36620673; PMCID: PMC9813515.
Author Response
Dear Reviewer, thank you for your comments. We have addressed all your concerns and changed the manuscript accordingly. Below you will find your original comments together with our answers and reference(s) to the point(s) changed the manuscript, in italics.
My major concerns are:
Reading throughout the paper it does not seems as a study protocol since, for instance, it is stated that: “Methods. The research uses a cross-sectional, quasi-experimental design with 40 children aged six to eight with ASD.”; … “It is expected that all scores will improve from pre-test to post-test in the experimental group”
Dear Reviewer, we modified the abstract adapting the sentences to the protocol writing. Thank you!
2.2.1. Inclusion Criteria: To be eligible for the study, children must (1) have an ASD diagnosis; (2) be between six and eight years old or have an equivalent mental age; (3) be capable of communicating orally; and (4) have sufficient motor skills.” I have some concerns regarding those criteria, particularly how to assess “equivalent mental age” and the “sufficient motor skills”?
“You are right. We wrote more detailed criteria about measures and skills for mental age, communication skills and motor skills, modifying this part as follow:
To be eligible for the study, children must meet following inclusion criteria: (1) to have received an ASD diagnosis; (2) to be between six and eight years old or have an equivalent mental age, calculated by the standardized test The Correspondences and Functions Evaluation (which corresponds to Italian C.F.V. – Corrispondenze e Funzioni Valutazione) test (Vianello e Marin, 1998): This test has 42 items, subdivided into five areas of logical operations: qualitative correspondences, direct quantitative correspondences, indirect quantitative correspondences, direct functions and indirect functions. Each item has a binary evaluation, with a mark of 1 for a correct answer and 0 for an incorrect answer. The raw data are transformed in mental age (3 – 14 yrs) based on appropriate conversion tables [32]; (3) to master language skills as early forms of understanding, listening and expressing; pointing to body parts; following instructions, listening and paying attention; use of names; sentence language; asking questions; and (4) to master motor skills including Fundamental motor skills (FMS) of standing alone and walking, crawling, running, walking up/down stairs and jumping, throwing and catching. Language and motor skills will be measured through the VABS 2 -Vineland Adaptive Behavior Scales (Sparrow, Cicchetti and Balla, 2005) [33]. VABS 2 is the "gold standard" test to measure adaptive behaviors for everyday independent life from birth to 90 years old in intellectual or developmental disabilities, including autism spectrum disorders (ASD). It’s a semi structured interview with the caregiver to measure skills in four areas: Communication (receptive, expressive and written language), Daily Living Skills (skills to take care of oneself, household and community skills), Socialization (social relationship, emotional and behavioral regulation, leisure activities) and Motor Skills (fine and gross motor skills).
The research protocol proposes an intervention based on a short form of EMP, which comprises twelve units of the original protocol. Are there adaptations? Did the authors test them with children with ASD? What was the rationale to choose those 12 units from the 30? Which of them?
Thank you for this question, which allows us to be more specific. At the moment, we are working on reducing the program's number of units in response to requests from practitioners who have seen how, in clinical contexts, a program of 30 units is too long and tends to tire children. This is a first adaptation, and at the same time, we are also working with other neurodevelopmental disorders (ADHD) for ex post comparisons.
The reduction from 30 to 12 activities considered the representativeness of the EFs and the nature of the motor exercises, i.e., gross coordination exercises. Therefore, according to Diamond's model, equal numbers were selected for the three basic EFs: inhibitory control, working memory, and attentional shifting.
The measures to ascertain the impact of the intervention involve two tests assessing working memory capacity and two tests assessing inhibitory control. Is the FE-PS 2-6 battery for assessing EFs in children is validated for children with ASD?
No, they are not validated for children with ASD. Still, we selected these measures because they are standardized for preschool children and their use is consolidated for studying programs that stimulate EFs in this age group. Therefore, since the equivalent age of the children who will participate in our study is the same, we thought it appropriate to select these instruments that we are using in other studies. This will also allow us to compare with other samples.
The activity will be repeated with variations to support cognitive function development”. What type/kind of variations? If this is a paper to explain a protocol, every step and situation must be described.
We added all activities and variations in the appendix A. Thank you!
Limitations are not properly acknowledged in the paper.
We acknowledged the limitations in a separated section. Thank you!
Reviewer 2 Report
Comments and Suggestions for Authors
Dear Authors, The idea that executive function can be developed through physical and motor activities is well-supported by numerous studies. Any research exploring this connection is valuable for intervention programs. However, I have some comments regarding your research design: 1. You should validate the reliability and validity of your assessment items. Since your sample consists of children with ASD, it's important to ensure that the tests are appropriate for this population. 2. Your inclusion criteria need to be more clearly defined. Comorbidity is often present in populations with ASD, so this should be taken into account. 3. You must ensure that the duration of the intervention program is sufficient for your population and your hypothesis. You will find my remarks in your text. Thank you for your work on this important topic.

Author Response
Dear reviewer 2, thank you for your comments. We have addressed all your concerns and changed the manuscript accordingly. Below you will find your original comments together with our answers and reference(s) to the point(s) changed the manuscript, in italics.
Six weeks is insufficient time for improvement. You need more than eight weeks, especially because the program is conducted twice weekly.
Dear Reviewer, we know that the frequency of our intervention falls below the recommended three times per week, as suggested by the literature. However, most group interventions, complementary to classic therapies such as psychomotor skills and speech therapy, occur twice a week, as children with ASD typically participate in numerous other activities throughout the week. We added this specific reason in the “limitations” section.
Please provide more specific details. What does it mean to be capable of communicating orally? How will you measure this ability?
Additionally, what does "sufficient motor skills" entail? How will you assess this?
Since your program is centered around motor skills, it is essential to establish a baseline for motor development.
Ammended: Thank you.
Is this battery adapted for children with ASD?
No, they are not validated for children with ASD. Still, we selected these measures because they are standardized for preschool children and their use is consolidated for studying programs that stimulate EFs in this age group. Therefore, since the equivalent age of the children who will participate in our study is the same, we thought it appropriate to select these instruments that we are using in other studies. This will also allow us to compare with other samples.
Is it adapted for ASD?
No, the Backword Word Span is not adapted for children with ASD. We used the same selection criterion as for other measures, i.e. the mental equivalent age
Round 2
Reviewer 1 Report
Comments and Suggestions for Authors
I appreciate the enormous effort that you have taken to improve the overall quality of the manuscript and to respond to the reviewers' questions and concerns. In my opinion the paper is now suitable for publication.
Author Response
Dear Reviewer, thank you so much!
Reviewer 2 Report
Comments and Suggestions for Authors
All my previous comments were taken into consideration.
Author Response
Dear Reviewer, thank you so much!